# PREVENTING REWARD HACKING WITH OCCUPANCY MEASURE REGULARIZATION

## ABSTRACT

Reward hacking occurs when an agent performs very well with respect to a specified or learned reward function (often called a "proxy"), but poorly with respect to the true desired reward function. Since ensuring good alignment between the proxy and the true reward is remarkably difficult, prior work has proposed regularizing to a "safe" policy using the KL divergence between action distributions. The challenge with this divergence measure is that a small change in action distribution at a single state can lead to potentially calamitous outcomes. Our insight is that when this happens, the state occupancy measure of the policy shifts significantly—the agent spends time in drastically different states than the safe policy does. We thus propose regularizing based on occupancy measure (OM) rather than action distribution. We show theoretically that there is a direct relationship between the returns of two policies under *any* reward function and their OM divergence, whereas no such relationship holds for their action distribution divergence. We then empirically find that OM regularization more effectively prevents reward hacking while allowing for performance improvement on top of the safe policy.

## 1 INTRODUCTION

A major challenge for the designers of goal-oriented AI systems is specifying a reward function that robustly captures their goals and values. When the specified reward is misaligned with the designer's intent, it is really just a *proxy* for the true reward. This can lead to a phenomenon called *reward hacking*: the resulting policy accumulates achieves high reward according to the proxy reward function, but not according to the true reward function (Russell et al., 2010; Amodei et al., 2016; Pan et al., 2022; Skalse et al., 2022). The resulting behavior is often undesirable and can be especially catastrophic when deploying these systems in safety-critical scenarios, such as autonomous driving (Krakovna et al., 2019; Turner et al., 2019; Knox et al., 2022).

The best solution to prevent reward hacking would be to ensure better alignment between the specified proxy and hidden true reward. However, in practice, reward functions are extremely difficult to properly design due to the ambiguities and complex variables underlying real-world scenarios (Ibarz et al., 2018). For example, recommender systems aim to optimize the value that users attain from their time spent on the online platforms, but since this goal is difficult to quantify, designers utilize proxies, such as click-through rates, engagement time, and other types of explicit feedback they receive from users, which do not always match how satisfied users implicitly are with their experience (Stray et al., 2022). Several examples of reward hacking have been reported throughout the literature (Kraknova & Legg, 2020).

Rather than specifying a reward function by hand, an alternative is to learn it from human feedback like demonstrations (Ho & Ermon, 2016), comparisons (Sadigh et al., 2017; Christiano et al., 2017), or both (Palan et al., 2019) However, even learned reward functions are often misaligned with our true objectives, failing to generalize outside the distribution of behavior used in training (McKinney et al., 2023; Tien et al., 2023).

Instead of blindly optimizing a proxy reward function, one proposal to avoid reward hacking is to *regularize the policy's chosen actions to be similar to those of a known safe policy* (Yang et al., 2021). For example, RLHF for LLMs generally optimizes the learned reward in addition to a term that penalizes divergences from the pre-trained language model's output (Glaese et al., 2022; Ouyang et al., 2022). Intuitively, this kind of regularization pushes the learned policy away from "unusual"

behaviors for which the reward function may be misaligned. These could include unforeseen strategies in the case of a hand-specified reward function or out-of-distribution states in the case of a learned reward function. Initializing using a pre-trained policy has been shown to effectively speed up the learning process (Laidlaw et al., 2023; Uchendu et al., 2023), and a similar paradigm can also be used to ensure the safety of the agent during online inference (Gulcehre et al., 2023).

However, regularizing based on the *action distributions* of policies has significant drawbacks. Small shifts in action distributions can lead to large differences in outcomes. And vice-versa, large shifts in action distributions may not actually result in any difference in outcome and thus wrongly interfere with training the policy.

Imagine an autonomous car driving along a coastal highway with a steep cliff to its right. For illustrative purpuses, let us think of the states as positions, and actions as velocities. Suppose we have access to a safe policy that drives slowly and avoids falling off the cliff, but the car is optimizing a proxy reward function that incentivizes progress rather than staying on the road. If we try to regularize the car's action distributions to the safe policy, we may have to apply a lot of regularization since one wrong action going too far to the right—a small change in the action distribution at a single state—can lead to disaster. This makes it very difficult for the regularizer to be successful. To make matters worse, optimizing the proxy will lead to moving faster, likely through action changes at many states, which makes going more to the right at a single state a negligible divergence. It is difficult if not impossible to avoid a single catastrophic action while improving at all on the safe policy.

Xu et al. (2020)

If action distributions induce poor regularizers for reward hacking, then what can we do instead? Thinking back to the car example, while the single catastrophic action didn't change the action distribution much, observe that it *did* change something quite drastically: *the resulting distribution over states visited by the car*. The learned policy will have a high probability of reaching states where the car is off the cliff and crashed, while the safe policy never reaches such states. *Our proposal follows naturally from this observation: to avoid reward hacking, regularize based on divergence from the safe policy's **occupancy measure**, rather than action distributions.* An occupancy measure (OM) represents the distribution of states (and optionally actions) seen by a policy when it interacts with its environment. Unlike action distribution-based metrics, OM takes into account not just the actions taken by the policy, but also the states that the agent reaches.

In this paper, we show both theoretically and empirically that regularizing policy optimization using occupancy measure divergence is more effective at preventing reward hacking. Theoretically, we prove that there is a direct relationship between the returns of two policies under *any* reward function and how much they deviate from each other in occupancy measure space, and that no such relationship holds for divergences between the action distributions of the two policies. Empirically, regularizing the occupancy measure of a policy is more challenging than regularizing its action distributions. Action distribution-based policy regularization implicitly assumes a distance metric over the space of policies. So far, this has been the Kullback–Leibler (KL) divergence, which has several attractive advantages, including the fact that it is easy to compute and optimize within common deep RL algorithms (Vieillard et al., 2021). To address this, we derive an algorithm called **Occupancy-Regularized Policy Optimization** (ORPO) that can also be easily incorporated into deep RL algorithms like Proximal Policy Optimization (PPO). ORPO approximates the occupancy measure divergence between policies using a discriminator network.

We use ORPO to optimize policies trained with misaligned proxy reward functions in multiple reward hacking benchmark environments (Pan et al., 2022) and compare our method's performance to that of action distribution regularization. The results of our experiments show that regularization based on occupancy measures more effectively prevents reward hacking while allowing performance improvement over the base policy. Our findings suggest that regularization based on occupancy measures should replace action distribution-based regularization to ensure the safety of goal-driven AI systems in the real world, while also allowing them to provide value.

## 2 RELATED WORK

Some prior works have focused on characterizing and defining theoretical models of reward hacking as a special case of Goodhart's Law (Goodhart, 1984; Krakovna, 2019; Skalse et al., 2022; Ngo et al.,

2023). Skalse et al. (2022) formally define reward hacking, also known as reward gaming (Leike et al., 2018), as an increase in a proxy reward function accompanied by a noticeable drop in the true reward function. Kraknova & Legg (2020) provide a list of many examples of reward hacking from the literature, but only a few studies have been conducted to understand the practical effects of reward hacking. Pan et al. (2022) systematically investigate reward misspecification and show that increasing the optimization power of RL agents can result in sudden shifts, or phase changes, in the agents' reward hacking behavior; they also define the reward hacking benchmark environments that we use to validate our method.

A few safety methods have been proposed to avoid reward hacking and/or mitigate its dangerous effects. An AI agent can be mildly optimized, such that they aim to achieve performance that is just "good enough" on the misspecified proxy reward function and not necessarily the absolute optimum (Taylor et al., 2020). Quantilizers proposed by Taylor (2016) is one such approach to avoid reward hacking behaviors by training agents that are mildly optimized; however, in practice, this isn't feasible to implement. Constrained reinforcement learning has been another safety method in which designers have tried to prevent the misbehavior of agents motivated by flawed reward functions (Dalal et al., 2018; Zhang et al., 2020; Chow et al., 2019); however, in general, these approaches are limited by the number of constraints that can be applied and the specification of weights that are to be applied to the constraints. In addition, their overly conservative approach to optimization restricts the agent from actually being helpful. Roy et al. (2022) try to overcome the problem of specifying weights for the constraints but still require the designer to specify the frequency of certain events, which is also difficult to do in practice.

Other proposals to address the reward specification problem attempt to infer the true reward function based on the given proxy reward function, environment context, and/or feedback from humans (Hadfield-Menell et al., 2017; Reddy et al., 2020). However, these approaches are limited due to the assumptions they make about the reward functions or the environment. Additionally, involving humans-in-the-loop (Lee et al., 2021) can be more expensive than providing offline demonstrations like the ones required by our method.

Regularizing policies to be similar to an offline policy based on their KL divergences was first proposed by Stiennon et al. (2020) and has since been widely employed in the context of optimizing LLMs using learned reward functions (Ouyang et al., 2022; Bai et al., 2022; Glaese et al., 2022). KL regularization for RLHF has been further studied by Vieillard et al. (2021), Gao et al. (2022), and Korbak et al. (2022). Different offline RL methods have also been proposed to ensure there isn't a distribution shift between a offline safe policy and the online learned policy (Kang et al., 2022; Ghosh et al., 2022). Vuong et al. (2022) uses discriminators like our algorithm to maintain closeness to the baseline; however, they use two discriminators to get a combination of reward maximization and regularization. Our method aims to do the same thing but only relies on one discriminator, and it is unique in its emphasis on solving reward hacking.

## 3    OCCUPANCY MEASURE-BASED REGULARIZATION

We formalize our policy regularization method in the setting of an infinite-horizon Markov decision process (MDP). An agent takes actions $a \in \mathcal{A}$ to transition between states $s \in \mathcal{S}$ over a series of timesteps $t = 0, 1, 2, \ldots$. The first state $s_0$ is sampled from an initial distribution $\mu_0(s)$, and when an agent takes action $a_t$ in $s_t$ at time $t$, the next state $s_{t+1}$ is reached at timestep $t + 1$ with transition probability $p(s_{t+1} \mid s_t, a_t)$. The agent aims to optimize a reward function $R : \mathcal{S} \times \mathcal{A} \to [0, 1]$, and rewards are accumulated over time with discount factor $\gamma \in [0, 1)$. A policy $\pi$ maps each state $s$ to a distribution over actions to take at that state $\pi(a \mid s)$. We define the (normalized) *return* of a policy $\pi$ under a reward function $R$ as

$$J(\pi, R) = (1 - \gamma) \mathbb{E}_\pi \left[ \sum_{t=0}^{\infty} \gamma^t R(s_t, a_t) \right]$$

where $\mathbb{E}_\pi$ refers to the expectation under the distribution of states and actions induced by running the policy $\pi$ in the environment. The normalizing factor guarantees that $J(\pi, R) \in [0, 1]$ always.

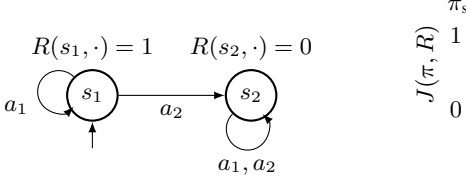

Figure 1: The MDP on the left, used in the proof of Proposition 3.1, demonstrates one drawback of using divergence between policies' action distributions for regularization. The agent stays in state $s_1$, where it receives 1 reward per timestep, until it takes action $a_2$, after which it remains in state $s_2$ forever and receives no reward. The plot on the right shows the return $J(\pi, R)$ for a policy $\pi$ when $\gamma = 0.99$ as a function of the policy's action distribution at $s_1$. While $\pi_{\text{safe}}$ and $\pi$ (shown on the plot as dotted lines) are close in action distribution space, they achieve very different returns. Meanwhile, $\pi$ is far from $\pi_{\text{safe}}'$ in action distribution space but achieves nearly the same return. Propositions 3.2 and A.2 show that occupancy measure divergences do not have these drawbacks.

We define the *state-action occupancy measure* $\mu_\pi$ of a policy $\pi$ as the expected discounted number of times the agent will be in a particular state and take a specific action:

$$\mu_\pi(s, a) = (1 - \gamma)\mathbb{E}_\pi\left[\sum_{t=0}^\infty \gamma^t \mathbb{1}\{s_t = s \wedge a_t = a\}\right].$$

Intuitively, the occupancy measure represents the distribution of states and actions visited by the policy over time. If $\pi$ spends a lot of time taking action $a$ in state $s$, then $\mu_\pi(s, a)$ will be high, whereas if $\pi$ never visits a state $s$, then $\mu_\pi(s, a) = 0$ for all actions $a$.

The standard approach to solving an MDP is to find a policy $\pi$ that maximizes its return:

$$\text{maximize} \quad J(\pi, R). \tag{1}$$

However, as we discussed in section 1, an AI system designer most likely does not have access to a reward function that perfectly encapsulates their preferences. Instead, the designer might optimize $\pi$ using a learned or hand-specified *proxy* reward function $\tilde{R}$ which is misaligned with the *true* reward function $R$. Blindly maximizing the proxy reward function could lead to reward hacking.

## 3.1 ACTION DISTRIBUTION-BASED REGULARIZATION

A widely-used approach to prevent reward hacking behaviors is to optimize the policy's return with respect to the proxy $\tilde{R}$ plus a regularization term that penalizes the KL divergence of the policy's action distribution from a *safe policy* $\pi_{\text{safe}}$:

$$\text{maximize} \quad J(\pi, \tilde{R}) - \lambda(1 - \gamma)\mathbb{E}_\pi\left[\sum_{t=0}^\infty \gamma^t D_{\text{KL}}(\pi(\cdot \mid s_t) \| \pi_{\text{safe}}(\cdot \mid s_t))\right]. \tag{2}$$

The regularization term can be easily incorporated into deep RL algorithms, like PPO, since the KL divergence between action distributions can usually be calculated in closed form.

Although it is simple, the action distribution-based regularization method in (2) has serious drawbacks that arise from the complex relationship between a policy's action distribution at various states and its return under the true reward function. In some cases, a very small change in action distribution space can result in a huge change in reward, and in other cases, a large change in action distribution space can result in a negligible change in reward. We formalize this in the following proposition.

**Proposition 3.1.** *Fix $\epsilon > 0$ and $\delta > 0$ arbitrarily small, and $c \geq 0$ arbitrarily large. Then there is an MDP and true reward function $R$ where both of the following hold:*

*1. There is a pair of policies $\pi$ and $\pi_{safe}$ where the action distribution KL divergence satisfies*

$$(1 - \gamma)\mathbb{E}_\pi\left[\sum_{t=0}^\infty \gamma^t D_{KL}(\pi(\cdot \mid s_t) \| \pi_{safe}(\cdot \mid s_t))\right] \leq \epsilon$$

*but $|J(\pi_{safe}, R) - J(\pi, R)| \geq 1 - \delta$.*

2. *There is a safe policy $\pi_{safe}{}'$ such that* any *other policy $\pi$ with*

$$(1 - \gamma)\, \mathbb{E}_\pi \left[ \sum_{t=0}^{\infty} \gamma^t D_{KL}(\pi(\cdot \mid s_t) \parallel \pi_{safe}{}'(\cdot \mid s_t)) \right] \leq c$$

*satisfies* $|J(\pi_{safe}{}', R) - J(\pi, R)| \leq \delta$.

The first part of Proposition 3.1 shows that in the worst case, a KL divergence below than some arbitrarily small threshold $\epsilon$ from the safe policy's action distributions can induce a change in the return under true reward function $R$ that is almost as large as the entire possible range of returns. Thus, when regularizing using action distribution KL divergence like in (2), one might have to make $\lambda$ extremely large to prevent drastic changes from the safe policy. However, the second part of Proposition 3.1 shows that in the same MDP, for a different safe policy, any learned policy with an arbitrarily large action distribution KL divergence from the safe policy has an extremely small difference in return. This means that one might need to set $\lambda$ extremely small in order to allow for the large divergence in the policies' action distributions necessary for optimization to have any effect. Since Proposition 3.1 suggests that we need to make $\lambda$ large for some reasons and small for others, it may be impossible to choose a good $\lambda$ value that both prevents undesirable behavior but still allows some optimization of the reward function. See Figure 1 for a graphical illustration of Proposition 3.1 and Appendix A.1 for a proof.

### 3.2 Occupancy measure-based regularization

Since Proposition 3.1 shows that small KL divergence in action space from the safe policy can have large effects, and vice versa, it may be impossible in some environments to regularize effectively using the objective in (2). We propose instead to regularize based on the divergence between the occupancy measures of the learned and safe policies:

$$\text{maximize} \quad J(\pi, \tilde{R}) - \lambda \left\| \mu_\pi - \mu_{\pi_{\text{safe}}} \right\|_1. \tag{3}$$

In (3), we use the total variation (TV) between the occupancy measures, defined as

$$\left\| \mu_\pi - \mu_{\pi_{\text{safe}}} \right\|_1 = \sum_{(s,a) \in \mathcal{S} \times \mathcal{A}} |\mu_\pi(s,a) - \mu_{\pi_{\text{safe}}}(s,a)|.$$

Why should using the occupancy measure divergence to regularize perform better than using the divergence between action distributions? The following proposition shows that the TV distance between occupancy measures does not have the same problems as action distribution divergence: a small divergence *cannot* result in a large change in policy return.

**Proposition 3.2.** *For any MDP, reward function $R$, and pair of policies $\pi, \pi_{safe}$, we have*

$$|J(\pi_{safe}, R) - J(\pi, R)| \leq \left\| \mu_\pi - \mu_{\pi_{safe}} \right\|_1. \tag{4}$$

Proposition 3.2 shows that, unlike action distribution divergences, the occupancy measure divergence between $\pi$ and $\pi_{\text{safe}}$ bounds the returns between the two policies. Specifically, a small difference between the occupancy measure of the safe and learned policies *guarantees* that they have similar returns. Even more, this bound is actually tight, suggesting that changes in occupancy measure TV distance will correspond closely with changes in the return of the policy under the true reward. Proposition A.2 in Appendix A shows that, in general, we cannot do any better than the bound from Proposition 3.2. See Appendix A.2 for a proof of this proposition.

These results suggest that the divergence between the occupancy measures of the learned and safe policies is a much better measure of how similar those policies are than the divergence between the policies' action distributions. In the following sections, we will show that these theoretical results match with intuition and empirical performance.

### 3.3 Example

Figure 2 shows an intuitive example of why regularizing to a safe policy based on occupancy measure divergence is superior to regularizing using action distribution divergence. Figure 2a depicts a

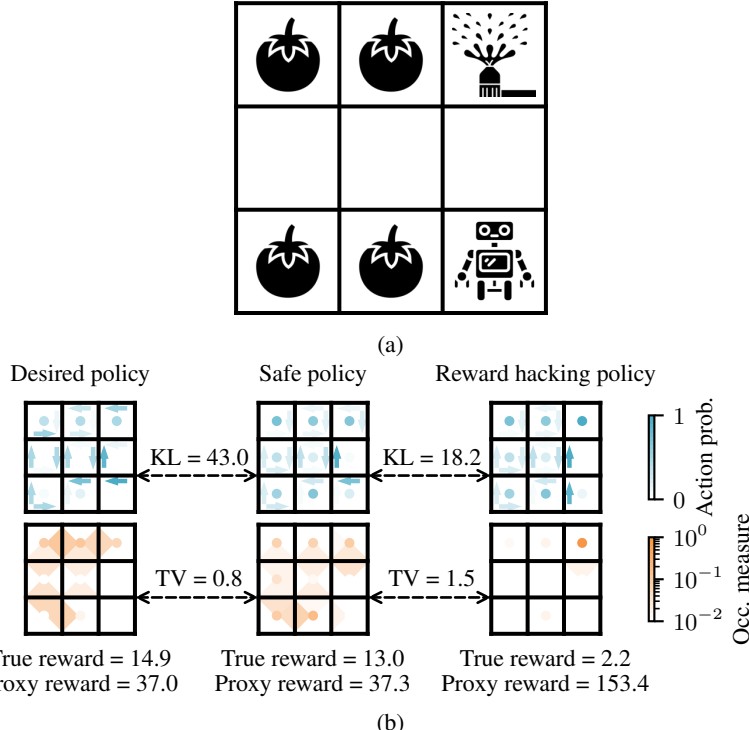

(a)

Desired policy     Safe policy     Reward hacking policy

KL = 43.0    KL = 18.2

TV = 0.8    TV = 1.5

True reward = 14.9    True reward = 13.0    True reward = 2.2
Proxy reward = 37.0    Proxy reward = 37.3    Proxy reward = 153.4

(b)

Figure 2: This simple gridworld provides an intuitive example of why occupancy measure divergences are superior to action distribution divergences for regularizing to a safe policy. See Section 3.3 for the details.

simplified version of the tomato-watering AI Safety Gridworld proposed by Leike et al. (2017). The agent, a robot that starts in the lower right corner, can move up, down, left, right, or stay in place. Its objective is to water the tomatoes on the board, and it receives reward each time it moves into a square with a tomato. However, there is also a sprinkler in the upper right corner of the environment. When the robot moves into the sprinkler's square, its sensors see water everywhere, and it believes all tomatoes are watered. In this environment, the true reward function $R$ only rewards watering tomatoes, while the proxy reward function $\tilde{R}$ also highly rewards reaching the sprinkler.

The top row of Figure 2b shows three policies for this environment: a desired policy that achieves the highest true reward, a safe policy that achieves lower true reward because it stays in place more often, and a reward hacking policy that exploits the sprinkler state to achieve high proxy reward but low true reward. The arrows between the policies on the top row of Figure 2b show the action distribution KL divergences between them as used for regularization in (2). The action distribution divergences suggest that the reward hacking policy is actually closer to the safe policy than the desired policy is. This is because the safe policy is nearly identical to the reward hacking policy in the upper right square, where the reward hacking policy spends most of its time; they both take the "stay" action with high probability. Thus, if we regularize to the safe policy using action distribution KL divergence, we would be more likely to find a policy that hacks the proxy reward, rather than one like the left policy, which we prefer. The problem is that the safe policy rarely reaches the sprinkler state, but the action distribution divergence doesn't account for this.

Using occupancy measure divergences avoids this problem. The bottom row of Figure 2b shows the occupancy measures for each policy in the top row, and the arrows between the columns show the total variation distance $\|\mu - \mu'\|_1$ between the policies' occupancy measures. In terms of occupancy measure distance, the desired policy on the left is closer to the safe policy than the reward hacking policy is. This is because both the desired and safe policies spend most of their time actually watering tomatoes, as evidenced by the higher occupancy measure they assign to the tiles on the board with the tomatoes. In contrast, the reward hacking policy spends almost all of its time in the sprinkler square and as a result, has a very different occupancy measure. Thus, if we trained a policy regularized with

occupancy measure divergence in this environment, we could hope to find a policy like the desired one on the left and avoid a reward hacking policy like the one on the right.

### 3.4 OCCUPANCY-REGULARIZED POLICY OPTIMIZATION (ORPO)

In the previous sections, we showed strong theoretical evidence that regularizing using occupancy measure divergences is superior to regularizing using action distribution divergences. We now introduce an algorithm, Occupancy-Regularized Policy Optimization (ORPO), to feasibly approximate the occupancy measure divergence between the learned and safe policies for the purpose of regularizing deep RL agents.

While our theory relies on the TV distance between occupancy measures, we find that the KL divergence is more stable to calculate in practice. Since Pinsker's inequality bounds the TV distance by the KL divergence for small KL values, and the Bretagnolle-Huber bound holds for larger KL values, our theoretical guarantees intuition remain valid in the case of OM KL (Canonne, 2022). Our objective from (3) can be reformulated with the KL divergence in place of the TV distance:

$$\text{maximize} \quad J(\pi, \tilde{R}) - \lambda\, D_{\text{KL}}(\mu_\pi \parallel \mu_{\pi_{\text{safe}}}). \tag{5}$$

We optimize (5) using a gradient-based method. The gradient of the first term is estimated using PPO, a popular policy gradient method (Schulman et al., 2017). However, calculating the occupancy measure divergence for the second term is intractable to do in closed form since it requires the enumeration of *all* possible state-action pairs, an impossible task in the case of deep RL. Thus, we approximate the KL divergence between the occupancy measures of policies by training a *discriminator network*, a technique that has previously been used for generative adversarial networks (GANs) (Goodfellow et al., 2014) and GAIL (Ho & Ermon, 2016).

The discriminator network $d : \mathcal{S} \times \mathcal{A} \to \mathbb{R}$ assigns a score $d(s, a) \in \mathbb{R}$ to any state-action pair $(s, a) \in \mathcal{S} \times \mathcal{A}$, and it is trained on a mixture of data from both the learned policy $\pi$ and safe policy $\pi_{\text{safe}}$. The objective used to train $d$ incentives low scores for state-action pairs from $\pi_{\text{safe}}$ and high scores for state-action pairs from $\pi$:

$$d = \arg\min_d \sum_{t=0}^{\infty} \left( \mathbb{E}_\pi [\, \gamma^t \log(1 + e^{-d(s_t, a_t)}) \,] + \mathbb{E}_{\pi_{\text{safe}}} [\, \gamma^t \log(1 + e^{d(s_t, a_t)}) \,] \right). \tag{6}$$

Huszár (2017) proves that if the loss function in (6) is minimized, then the expected discriminator scores for state-action pairs drawn from the learned policy distribution will approximately equal the KL divergence between the occupancy measures of the two policies:

$$D_{\text{KL}}(\mu_\pi(s, a) \parallel \mu_{\pi_{\text{safe}}}(s, a)) \approx (1 - \gamma)\mathbb{E}_\pi \left[ \sum_{t=0}^{\infty} \gamma^t d(s_t, a_t) \right]$$

Applying the definitions of the learned policy returns and the KL divergence between the polices' occupancy measures, we can now rewrite our ORPO objective:

$$\text{maximize} \quad \mathbb{E}_\pi \left[ \sum_{t=0}^{\infty} \gamma^t \left( \tilde{R}(s_t, a_t) - \lambda\, d(s_t, a_t) \right) \right]. \tag{7}$$

This objective dynamically changes at every step of the optimization–as the policy changes online, the discriminator will also adapt and change.

Note that (7) is identical to the normal RL objective with a reward function $R'(s, a) = \tilde{R}(s, a) - \lambda d(s, a)$. Thus, once the discriminator has been trained, we add the discriminator scores to the given reward function and use the combined values to update $\pi$ with PPO. The training process for ORPO consists of iterating between two phases: one in which data from both the safe and current policies is used to train the discriminator to minimize (6), and one in which data from the current policy is used to train the PPO agent with the augmented reward function in (7). After a policy gradient step, the process repeats.

| Method | Environment | | |
|---|---|---|---|
| | Tomato | Traffic | Glucose |
| $\pi_{\text{safe}}$ | $65.3 \pm 0.0$ | $-2296 \pm 0.0$ | $-76199 \pm 0.0$ |
| No regularization | $22.2 \pm 7.0$ | $-56425 \pm 6877.4$ | $-577093 \pm 33735.3$ |
| Action dist. regularization | $67.1 \pm 3.9$ | $-1212 \pm 24.2$ | $-75955 \pm 2604.0$ |
| State occupancy regularization | $66.7 \pm 0.8$ | $-1244 \pm 242.7$ | $-54287 \pm 27650.2$ |
| State-action occupancy regularization | $\mathbf{69.1} \pm 13.1$ | $\mathbf{-1096} \pm 36.2$ | $\mathbf{-14834} \pm 25365.3$ |

Table 1: The true rewards achieved by ORPO and PPO with action distribution regularization in the three reward hacking environments. We report the best reward achieved across a range of coefficients and compare to the safe policy $\pi_{\text{safe}}$ and training without regularization on the proxy reward.

While we have thus far considered the state-action occupancy measure of a policy $\mu_\pi(s, a)$, we find that in practice it sometimes makes more sense to regularize based on the state-only occupancy measure $\mu_\pi(s) = (1 - \gamma)\mathbb{E}_\pi[\sum_{t=0}^{\infty} \gamma^t \mathbb{1}\{s_t = s\}]$. In particular, in many environments the reward function $R(s)$ is only a function of the state. In this case, it is simple to establish similar guarantees to Proposition 3.2 based on the state-only occupancy measure, and so the state occupancy measure might be more effective in regularizing the behavior of the agent. We can implement this within ORPO by only providing the state as input to the discriminator rather than a state-action pair.

## 4 EXPERIMENTS

We consider the empirical performance of ORPO in four environments: the tomato-watering gridworld we focused on in Section 3.3; Flow, an autonomous vehicle control environment introduced by Wu et al. (2022); SimGlucose, and a blood glucose monitoring system developed by Fox et al. (2020). We chose the first for illustrative purposes, and the following two because they were presented as reward hacking benchmark environments by Pan et al. (2022).

**Tomato Environment**: As before, the tomato environment contains a sprinkler state where the agent perceives all tomatoes as being watered and thus receives high proxy reward but no true reward. For our safe policy, we train a PPO agent with the true reward function, and then add a 15% chance of taking a random action to ensure there is room to improve upon it.

**Flow Traffic Simulator**: The traffic environment simulates a road network where cars on an on-ramp attempt to merge into traffic on a highway. Some vehicles are controlled by a human model and some are RL-controlled autonomous vehicles. The true objective of the self-driving agent is to ensure that there is fast traffic flow at all times in order to reduce the mean commute time, while observing the positions and velocities of nearby vehicles. The proxy reward is the average velocity of all cars in the simulation. When the traffic agent begins to reward hack, it stops cars on the on-ramp from merging into traffic. This way, the proxy reward is optimized because cars on the straightway can continue forward at a fast speed instead of having to wait for a car to merge, which increases the average velocity of all vehicles. However, the true objective is not achieved as the commute time for the cars on the on-ramp increases indefinitely. As the safe policy for the traffic environment we used the Intelligent Driver Model (IDM), a standard approximation of human driving behavior (Treiber et al., 2000). In practice, safe policies are often learned via imitation learning, so to simulate this we generate data from the IDM controller and train a behavioral cloning (BC) policy on it with some entropy added to ensure that the policy more closely resembles the quality of baseline that can be reasonably provided in practice.

**SimGlucose**: This blood glucose monitoring environment is an extension of the FDA-approved glucose monitoring simulator proposed by Man et al. (2014) for Type 1 Diabetes patients. The patient eats meals, while wearing a continuous glucose monitoring (CGM) device that makes noisy readings of the patient's blood glucose. The RL agent takes the action of administering insulin to the patient in order to maintain healthy glucose levels, while observing their history of CGM readings in addition to the previous amounts of insulin that it has delivered. The true reward is the health risk that the patient encounters given different amounts of insulin they receive, but the proxy reward prioritizes the monetary cost of the treatment incurred. As the safe baseline policy, we train a BC policy based

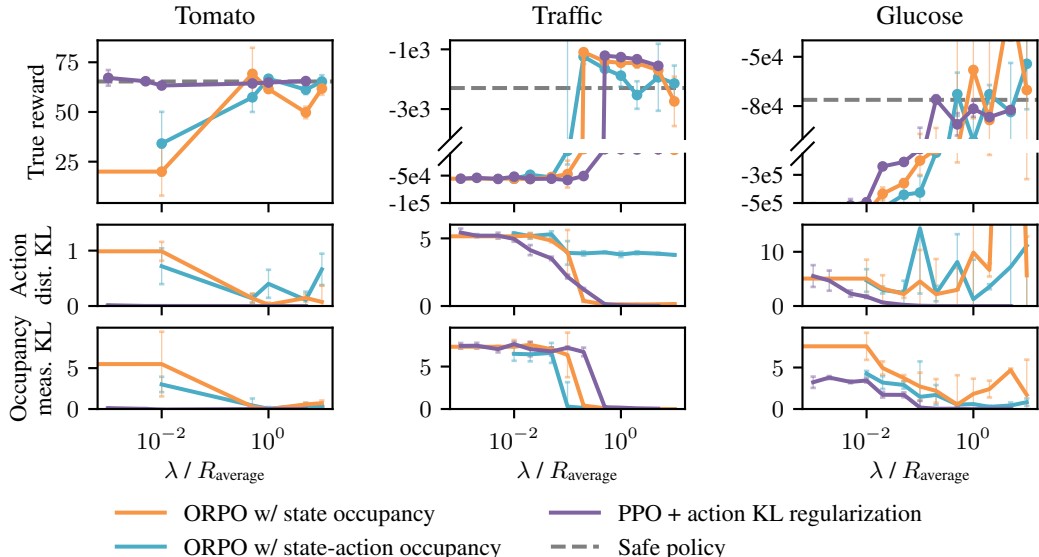

Figure 3: The results of running ORPO vs. PPO with action distribution regularization across a range of coefficients $\lambda$ in each of the three environments. We find that generally the regularization methods succeed at reducing the action distribution or occupancy measure KL to near-0 for high enough values of $\lambda$.

on actions given by a widely-used PID controller insulin pump with parameters tuned by the original designers of this system (Steil, 2013).

We train RL policies initialized from $\pi_{\text{safe}}$ in each environment using action distribution regularization and OM regularization, varying the regularization coefficient $\lambda$ across a wide range. In environments that we studied, $\lambda$ values between $10^-2$ and $10^2$ seemed to work best, scaled by their average per time step rewards; since the environments have vastly different magnitudes in reward, and the coefficients are applied to the per time step rewards, we scale coefficients in the range that we found worked best by multiplying by the average per time step reward. We compare the performance of the regularization techniques to the safe policies $\pi_{\text{safe}}$ to see if they can improve on $\pi_{\text{safe}}$ according to the true reward without devolving into reward hacking.

The results of our experiments are shown in Table 1 and Figure 3. In Table 1, we see that ORPO achieves higher true reward across all three environments compared to action distribution regularization. Figure 3 shows how the true reward, occupancy measure KL divergence, and action distribution KL divergence vary over values of the regularization coefficient $\lambda$. We see that as $\lambda$ is increased, the KL divergence eventually falls to 0 and the policy return approaches that of $\pi_{\text{safe}}$. Interestingly, we note that in general the optimal true reward is achieved at the value of $\lambda$ below which the occupancy measure KL suddenly increases. This observation could enable effective tuning of $\lambda$ in new environments, since we cannot always rely on having a true reward function to evaluate different values of $\lambda$ with.

## 5 CONCLUSION

We have presented theoretical and empirical evidence that occupancy measure regularization can more effectively prevent reward hacking than action distribution regularization when training with a misaligned proxy reward function. To address the practical challenges of the OM-based approach, we introduced an algorithm called ORPO, which uses an adversarially trained discriminator to approximate the KL divergence between the occupancy measures of policies. In the future, we hope to experiment with learned proxy reward functions in addition to the hand-specified reward functions we considered in this paper.

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

# APPENDIX

## A PROOFS

### A.1 PROOF OF PROPOSITION 3.1

**Proposition 3.1.** *Fix $\epsilon > 0$ and $\delta > 0$ arbitrarily small, and $c \geq 0$ arbitrarily large. Then there is an MDP and true reward function $R$ where both of the following hold:*

1. *There is a pair of policies $\pi$ and $\pi_{safe}$ where the action distribution KL divergence satisfies*

$$(1 - \gamma)\, \mathbb{E}_\pi \left[ \sum_{t=0}^\infty \gamma^t D_{KL}(\pi(\cdot \mid s_t) \parallel \pi_{safe}(\cdot \mid s_t)) \right] \leq \epsilon$$

   *but $|J(\pi_{safe}, R) - J(\pi, R)| \geq 1 - \delta$.*

2. *There is a safe policy $\pi_{safe}'$ such that* any *other policy $\pi$ with*

$$(1 - \gamma)\, \mathbb{E}_\pi \left[ \sum_{t=0}^\infty \gamma^t D_{KL}(\pi(\cdot \mid s_t) \parallel \pi_{safe}'(\cdot \mid s_t)) \right] \leq c$$

   *satisfies $|J(\pi_{safe}', R) - J(\pi, R)| \leq \delta$.*

*Proof.* Consider the following MDP, also shown in Figure 1:

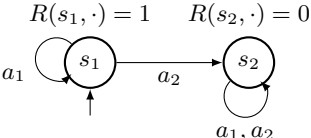

In this MDP, $\mathcal{S} = \{s_1, s_2\}$, $\mathcal{A} = \{a_1, a_2\}$, and the transition probabilities and reward function are defined by

$$\begin{aligned}
p(s_1 \mid s_1, a_1) &= 1 & p(s_2 \mid s_1, a_2) &= 1 \\
p(s_2 \mid s_2, a_1) &= 1 & p(s_2 \mid s_2, a_2) &= 1 \\
\forall a \in \mathcal{A} \quad R(s_1, a) &= 1 & R(s_2, a) &= 0.
\end{aligned}$$

The initial state is always $s_1$. Thus, the agent stays in state $s_1$ and receives 1 reward each timestep until it takes action $a_2$, at which point it transitions to $s_2$ and receives no more reward. Define for any $p \in [0, 1]$ a policy $\pi_p$ that takes action $a_2$ in $s_1$ with probability $p$, i.e. $\pi_p(a_2 \mid s_1) = p$; in $s_2$, suppose $\pi_p$ chooses uniformly at random between $a_1$ and $a_2$. Then

$$\begin{aligned}
J(\pi_p, R) &= (1 - \gamma) \sum_{t=0}^\infty \gamma^t \mathbb{P}(s_t = s_1) \\
&\overset{\text{(i)}}{=} (1 - \gamma) \sum_{t=0}^\infty \gamma^t (1 - p)^t \\
&\overset{\text{(ii)}}{=} \frac{1 - \gamma}{1 - \gamma(1 - p)}
\end{aligned} \tag{8}$$

where (i) is due to the fact that remaining in $s_1$ after $t$ timesteps requires $t$ independent events of $1 - p$ probability, and (ii) uses the formula for sum of an infinite geometric series.

We will prove the proposition using

$$\gamma = \max\left\{1 - \frac{\epsilon\delta}{2\log(2/\delta)}, 1 - \frac{\delta}{2}\right\}$$

$$\pi = \pi_{2(1-\gamma)/\delta}$$

$$\pi_{\text{safe}} = \pi_{(1-\gamma)\delta/2}$$

$$\pi_{\text{safe}}' = \pi_q \quad \text{where} \quad q = \max\left\{1 - \frac{1}{2\exp\{2(1/e + c(\delta + \gamma)/\delta)\}}, (1-\gamma)/\delta\right\}.$$

To start, we need to show that

$$(1-\gamma)\,\mathbb{E}_\pi\left[\sum_{t=0}^{\infty}\gamma^t D_{\text{KL}}(\pi(\cdot\mid s_t)\,\|\,\pi_{\text{safe}}(\cdot\mid s_t))\right] \le \epsilon. \tag{9}$$

Since $\pi$ and $\pi_{\text{safe}}$ are identical at $s_2$, we need only consider the KL divergence between the policies' action distributions at $s_1$. Thus we can rewrite the LHS of (9) as

$$(1-\gamma)\,\mathbb{E}_\pi\left[\sum_{t=0}^{\infty}\gamma^t D_{\text{KL}}(\pi(\cdot\mid s_t)\,\|\,\pi_{\text{safe}}(\cdot\mid s_t))\right] = (1-\gamma)\sum_{t=0}^{\infty}\gamma^t(1 - 2(1-\gamma)/\delta)^t D_{\text{KL}}(\pi(\cdot\mid s_1)\,\|\,\pi_{\text{safe}}(\cdot\mid s_1))$$

$$= \frac{\delta}{2\gamma + \delta}D_{\text{KL}}(\pi(\cdot\mid s_1)\,\|\,\pi_{\text{safe}}(\cdot\mid s_1))$$

$$\overset{(i)}{\le} \frac{\delta}{2}D_{\text{KL}}(\pi(\cdot\mid s_1)\,\|\,\pi_{\text{safe}}(\cdot\mid s_1)).$$

(i) is due to the fact that $\gamma \ge 1 - \delta/2$ by definition. Expanding the KL term gives

$$\frac{\delta}{2}\left(2(1-\gamma)/\delta\log\left(\frac{2(1-\gamma)/\delta}{(1-\gamma)\delta/2}\right) + \left(1 - 2(1-\gamma)/\delta\right)\log\left(\frac{1 - 2(1-\gamma)/\delta}{1 - (1-\gamma)\delta/2}\right)\right). \tag{10}$$

Assuming $\delta < 1$ (otherwise the result is trivially true), we have

$$2(1-\gamma)/\delta > (1-\gamma)\delta/2$$

$$1 - 2(1-\gamma)/\delta < 1 - (1-\gamma)\delta/2.$$

This implies that the right log term in (10) is negative, so we can bound (10) as

$$< (1-\gamma)\log\left(\frac{2(1-\gamma)/\delta}{(1-\gamma)\delta/2}\right)$$

$$= 2(1-\gamma)\log\left(\frac{2}{\delta}\right)$$

$$\overset{(i)}{\le} 2\frac{\epsilon\delta}{2\log(2/\delta)}\log\left(\frac{2}{\delta}\right)$$

$$= \epsilon,$$

which is the desired bound in (9). (i) uses the fact that $\gamma \ge 1 - \frac{\epsilon\delta}{2\log(2/\delta)}$ by definition.

Next, we will show that $|J(\pi_{\text{safe}}, R) - J(\pi, R)| \ge 1 - \delta$. First, we can calculate the return of $\pi_{\text{safe}}$ using (8):

$$J(\pi_{\text{safe}}, R) = \frac{1-\gamma}{1 - \gamma(1 - (1-\gamma)\delta/2)}$$

$$= \frac{1}{1 + \gamma\delta/2}$$

$$\overset{(i)}{\ge} 1 - \gamma\delta/2$$

$$\ge 1 - \delta/2. \tag{11}$$

(i) uses the fact that $\frac{1}{1+x} \geq 1 - x$ for positive $x$. The return of $\pi$ can be calculated similarly as

$$
\begin{aligned}
J(\pi, R) &= \frac{1 - \gamma}{1 - \gamma(1 - 2(1 - \gamma)/\delta)} \\
&= \frac{\delta}{2\gamma + \delta} \\
&\overset{(i)}{\leq} \frac{\delta}{2},
\end{aligned}
\tag{12}
$$

where (i) uses the fact that $\gamma \geq 1 - \delta/2$. Combining (11) and (12) gives $|J(\pi_{\text{safe}}, R) - J(\pi, R)| \geq 1 - \delta$ as desired.

To prove part 2, consider any $\pi$ satisfying

$$
(1 - \gamma) \mathbb{E}_\pi \left[ \sum_{t=0}^\infty \gamma^t D_{\text{KL}}(\pi(\cdot \mid s_t) \parallel \pi_{\text{safe}}{}'(\cdot \mid s_t)) \right] \leq c.
$$

Let $p = \pi(a_2 \mid s_1)$. Then clearly by the definition of $\pi_{\text{safe}}$,

$$
(1 - \gamma) \mathbb{E}_\pi \left[ \sum_{t=0}^\infty \gamma^t D_{\text{KL}}(\pi_p(\cdot \mid s_t) \parallel \pi_{\text{safe}}{}'(\cdot \mid s_t)) \right] \leq c,
\tag{13}
$$

i.e. $\pi_p$ also satisfies the inequality. Furthermore, note that $J(\pi, R) = J(\pi_p, R)$. We will show that $p \geq (1 - \gamma)/\delta$. This will imply that

$$
\begin{aligned}
J(\pi, R) = J(\pi_p, R) \\
= \frac{1 - \gamma}{1 - \gamma(1 - p)} \\
\leq \frac{1 - \gamma}{1 - \gamma(1 - (1 - \gamma)/\delta)} \\
= \frac{\delta}{\gamma + \delta} \qquad\qquad\qquad \leq \delta.
\end{aligned}
$$

Since $\pi_{\text{safe}}{}' = \pi_q$ and $q \geq (1 - \gamma)/\delta$ by definition, $J(\pi_{\text{safe}}{}', R) \leq \delta$ also. Since the return of both policies must also be nonnegative, this implies $|J(\pi_{\text{safe}}{}', R) - J(pi, R)| \leq \delta$, which is the desired bound.

Now, we just need to show that $p \geq (1 - \gamma)/\delta$. We do so by contradiction, i.e. assume that $p < (1 - \gamma)/\delta$. We can rewrite the LHS of (13) as

$$
\underbrace{\frac{1 - \gamma}{1 - \gamma(1 - p)}}_{(a)} \left[ \underbrace{p \log \left( \frac{p}{q} \right)}_{(b)} + \underbrace{(1 - p) \log \left( \frac{1 - p}{1 - q} \right)}_{(c)} \right].
\tag{14}
$$

We will give lower bounds for each part of (14). For (a), we have

$$
\frac{1 - \gamma}{1 - \gamma(1 - p)} > \frac{1 - \gamma}{1 - \gamma(1 - (1 - \gamma)/\delta)} = \frac{\delta}{\gamma + \delta}.
$$

For (b), note that $q \leq 1$, so

$$
p \log \left( \frac{p}{q} \right) \geq p \log p \geq -\frac{1}{e},
$$

since the function $f(x) = x \log x$ has its minimum at $f(x) = -1/e$. For (c), note that $1 - p > 1 - (1 - \gamma)/\delta \geq 1 - (1 - (1 - \delta/2))/\delta = 1/2$. Thus we can bound

$$
\begin{aligned}
(1 - p) \log \left( \frac{1 - p}{1 - q} \right) &> \frac{1}{2} \log \left( \frac{1}{2(1 - q)} \right) \\
&\geq \frac{1}{2} \log \left( \frac{1}{2 \frac{1}{2 \exp\{2(1/e + c(\delta + \gamma)/\delta)\}}} \right) \\
&= \frac{1}{e} + c \frac{\delta + \gamma}{\delta}.
\end{aligned}
$$

Combining the three bounds on the components of (14) gives

$$
(1-\gamma)\,\mathbb{E}_\pi\left[\sum_{t=0}^\infty \gamma^t D_{\mathrm{KL}}(\pi_p(\cdot \mid s_t) \parallel \pi_{\mathrm{safe}}'(\cdot \mid s_t))\right]
$$
$$
> \frac{\delta}{\gamma+\delta}\left[-\frac{1}{e}+\frac{1}{e}+c\frac{\delta+\gamma}{\delta}\right]
$$
$$
= c,
$$

which contradicts (13), thus completing the proof. $\square$

### A.2 PROOF OF PROPOSITION 3.2

We first prove another useful proposition:

**Proposition A.1.** *The return of a policy $\pi$ under a reward function $R$ is given by*

$$
J(\pi, R) = \sum_{(s,a)\in\mathcal{S}\times\mathcal{A}} \mu_\pi(s,a)R(s,a).
$$

*Proof.* Applying the definitions of return and occupancy measure, we have

$$
J(\pi, R) = (1-\gamma)\,\mathbb{E}_\pi\left[\sum_{t=0}^\infty \gamma^t R(s_t, a_t)\right]
$$
$$
= (1-\gamma)\sum_{t=0}^\infty \gamma^t \sum_{(s,a)\in\mathcal{S}\times\mathcal{A}} R(s,a)\,\mathbb{P}_\pi\left(s_t = s \wedge a_t = a\right)
$$
$$
= (1-\gamma)\sum_{(s,a)\in\mathcal{S}\times\mathcal{A}} R(s,a)\sum_{t=0}^\infty \gamma^t\,\mathbb{P}_\pi\left(s_t = s \wedge a_t = a\right)
$$
$$
= \sum_{(s,a)\in\mathcal{S}\times\mathcal{A}} R(s,a)\,(1-\gamma)\,\mathbb{E}_\pi\left[\sum_{t=0}^\infty \gamma^t \mathbb{1}\left\{s_t = s \wedge a_t = a\right\}\right]
$$
$$
= \sum_{(s,a)\in\mathcal{S}\times\mathcal{A}} \mu_\pi(s,a)R(s,a).
$$

$\square$

According to Proposition A.1, the return of a policy is simply a weighted sum of the reward function, where the weights are given by the occupancy measure. We now prove Proposition 3.2.

**Proposition 3.2.** *For any MDP, reward function $R$, and pair of policies $\pi, \pi_{safe}$, we have*

$$
|J(\pi_{safe}, R) - J(\pi, R)| \leq \left\|\mu_\pi - \mu_{\pi_{safe}}\right\|_1. \tag{4}
$$

*Proof.* Applying Proposition A.1, Hölder's inequality, and the fact that $R(s,a) \in [0,1]$, we have

$$
|J(\pi_{\mathrm{safe}}, R) - J(\pi, R)|
$$
$$
= \left|\sum_{(s,a)\in\mathcal{S}\times\mathcal{A}} (\mu_{\pi_{\mathrm{safe}}}(s,a) - \mu_\pi(s,a))R(s,a)\right|
$$
$$
\leq \left(\max_{(s,a)\in\mathcal{S}\times\mathcal{A}} |R(s,a)|\right)\left(\sum_{(s,a)\in\mathcal{S}\times\mathcal{A}} |\mu_{\pi_{\mathrm{safe}}}(s,a) - \mu_\pi(s,a)|\right)
$$
$$
\leq \left\|\mu_\pi - \mu_{\pi_{\mathrm{safe}}}\right\|_1.
$$

$\square$

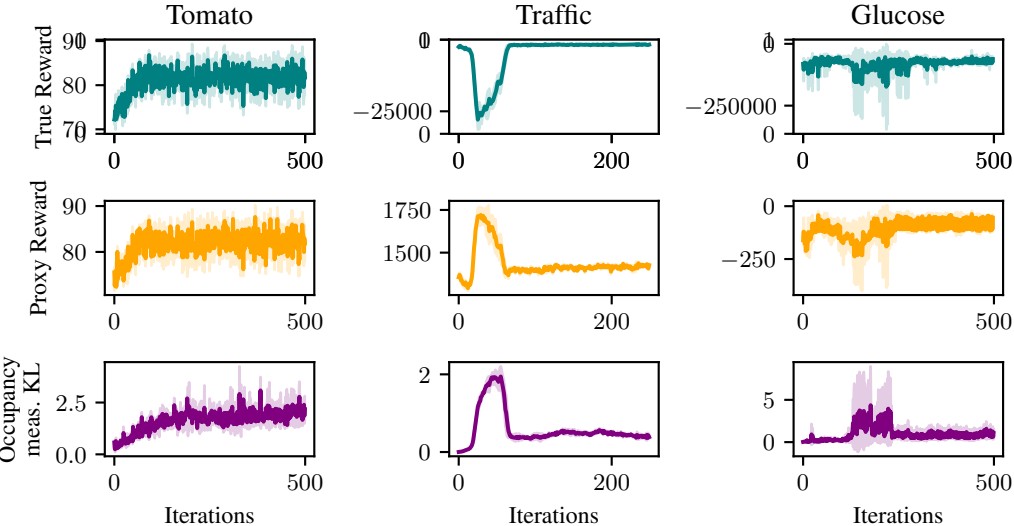

Figure 4: This figure shows training runs of ORPO in the three environments considered for experimentation. Note that we picked random $\lambda$ values to showcase in this graph, not just the best ones.

### A.3 ADDITIONAL RESULTS

The following proposition demonstrates that there is always some reward function for which the bound in (4) is tight up to a factor of two.

**Proposition A.2.** *Fix an MDP and pair of policies $\pi, \pi_{safe}$. Then there is some reward function $R$ such that*

$$|J(\pi_{safe}, R) - J(\pi, R)| \geq \frac{1}{2} \left\| \mu_\pi - \mu_{\pi_{safe}} \right\|_1.$$

*Proof.* Define two reward functions

$$R_1(s, a) = \mathbb{1}\{\mu_{\pi_{safe}}(s, a) \geq \mu_\pi(s, a)\}$$
$$R_2(s, a) = \mathbb{1}\{\mu_{\pi_{safe}}(s, a) \leq \mu_\pi(s, a)\}.$$

Using Proposition A.1, we have

$$
\begin{aligned}
&|J(\pi_{safe}, R_1) - J(\pi, R_1)| + |J(\pi, R_2) - J(\pi_{safe}, R_2)| \\
&\geq J(\pi_{safe}, R_1) - J(\pi, R_1) + J(\pi, R_2) - J(\pi_{safe}, R_2) \\
&= \sum_{(s,a) \in \mathcal{S} \times \mathcal{A}} \left( \mu_{\pi_{safe}}(s, a) - \mu_\pi(s, a) \right) \left( R_1(s, a) - R_2(s, a) \right) \\
&= \sum_{(s,a) \in \mathcal{S} \times \mathcal{A}} \left( \mu_{\pi_{safe}}(s, a) - \mu_\pi(s, a) \right) \begin{cases} 1 & \mu_{\pi_{safe}}(s, a) > \mu_\pi(s, a) \\ -1 & \mu_{\pi_{safe}}(s, a) < \mu_\pi(s, a) \\ 0 & \mu_{\pi_{safe}}(s, a) = \mu_\pi(s, a) \end{cases} \\
&= \sum_{(s,a) \in \mathcal{S} \times \mathcal{A}} \left| \mu_{\pi_{safe}}(s, a) - \mu_\pi(s, a) \right| \\
&= \left\| \mu_\pi - \mu_{\pi_{safe}} \right\|_1.
\end{aligned}
$$

Since both of the terms on the first line are positive, one must be at least $\frac{1}{2} \left\| \mu_\pi - \mu_{\pi_{safe}} \right\|_1$, which completes the proof. $\square$

## B  STABILITY AND CONVERGENCE OF ORPO

As we can see in 4, ORPO is generally stable, especially towards the end of the training run. It tends to converge to be within a narrow range for each of the metrics. Note that some of the unstable behavior that is shown in the graph can be because RL is in general a bit unstable to train. We do sometimes see instability at high values of $\lambda$ because the added rewards from the discriminator are very large compared to the underlying rewards and are changing over time. As we show in 3, ORPO works best with moderate values of $\lambda$ since moderate regularization both prevents reward hacking and allows improvement upon the provided safe policy. Thus, we do not believe that instability with high values of $\lambda$ presents a challenge to the practical adoption of our regularization method. It's also worth noting that in the case of the simple tomato environment, initializing with the safe policy already seems to keep the true and proxy reward functions well-aligned with each other in some cases.

## C  ENVIRONMENT DETAILS

### C.1  TOMATO ENVIRONMENT

In Figure 5, we have the setup of the tomato environment board we used for training.

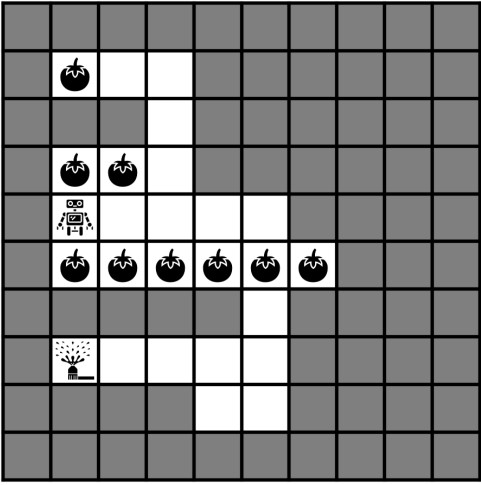

Figure 5: Here, the gray squares represent walls, and the white squares represent open spaces where the agent can travel.

The sprinkler state is down a shallow hallway, and on the other end a tomato is down another shallow hallway. We wanted to try out a scenario where the reward hacking would be relatively difficult for the agent to find to see whether or not our method works for more complex gridworld scenarios.

### C.2  TRAFFIC ENVIRONMENT

In Figure 6, we have a simplified rendering of the traffic flow environment merge scenario.

Within this particular frame, reward hacking is taking place. As we can see the blue RL vehicle has stopped completely on the on-ramp, resulting in cars to collect behind it. This way, the proxy reward, which is the average velocity of all vehicles in the simulation, is optimized as the cars on the straightway are able to continue speeding along the road without having to wait for merging cars. However, little to no true reward of the average commute time is achieved as the cars on the on-ramp aren't able to continue their commute.

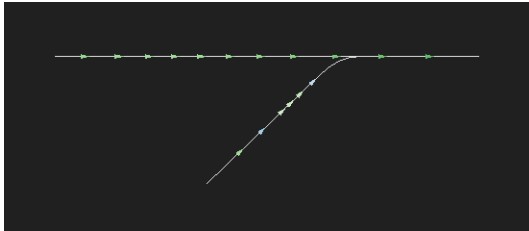

Figure 6: Here, the green cars are controlled by the human driver model IDM controller, and the blue cars are controlled by RL.

## D  EXPERIMENT DETAILS

Here, we give some extra details about the architectures and hyperparameters we used for training the ORPO agents. We build ORPO using RLLib (Liang et al., 2018) and PyTorch (Paszke et al., 2019). For all RL experiments we train with 3 random seeds and report the median reward.

**Network architectures**   The policy model for both the traffic and tomato environments was a simple fully connected network (FC-net) with a width of 512 and depth of 4. The policy model for the glucose environment is a basic LSTM network with 3 layers, each with widths of 64. We made this choice since the observation of the environment contains continuous historical information about the patient's blood glucose levels and previously administered insulin. The model sizes were chosen as we found that models with these capacities empowered the agents significantly enough for them to reward hack consistently.

The discriminator model for the tomato and traffic environments was a simple FC-net with a width of 256 and depth of 4. For the glucose environment, we defined multiple configurations for the discriminator due to the continuous nature of its observation space. First, we have an option to allow for the entire history of the patient that is captured in the observation by default to be fed into the discriminator network, in which case the discriminator will be an LSTM network similar to the policy network in order to properly handle the time series data. By default, the last four hours of the patient's state split into five minute intervals will be fed into the discriminator, but there is also an option to decrease the amount of history being used. If no history is used for the input to the discriminator network, we default to using the same FC-net used for the tomato and traffic environments. We additionally have the option of using the entire observation provided in the glucose environment (the CGM readings of the patient and the amount of insulin delivered) or just the CGM readings. Please refer to Section **??** for a discussion of why training the discriminator with different inputs can make a difference in how the policy is regularized.

### D.1  HYPERPARAMETERS

Some hyperparameters for the traffic environment were tuned by Pan et al. (2022).

The coefficient $\lambda$ that is used for determining how much regularization to apply was varied throughout the experiments and noted in our result.

| Hyperparameter | Value (Tomato) | Value (Traffic) |
|---|---|---|
| Training iterations | 500 | 250 |
| Batch size | 3000 | 6000 |
| | | |
| SGD minibatch size | 128 | 6000 |
| SGD epochs per iteration | 5 | 5 |
| Optimizer | Adam | Adam |
| Learning rate | 1e-3 | 5e-5 |
| Gradient clipping | 0.1 | 0.1 |
| Discount rate ($\gamma$) | 0.99 | 0.99 |
| GAE coefficient ($\lambda$) | 0.98 | 0.97 |
| Entropy coefficient | 0.01 | 0.01 |
| KL target | 0.01 | 0.02 |
| Value function loss clipping | 10 | 10,000 |
| Value function loss coefficient | 0.1 | 0.5 |

Table 2: PPO/ORPO hyperparameters.

| Hyperparameter | Value (Tomato) | Value (Traffic) |
|---|---|---|
| Discriminator reward clipping | 1000 | 10 |
| Regularization coefficient ($\lambda$) | Varied | Varied |

Table 3: ORPO-specific hyperparameters.

