# OpenReview forum: "Preventing Reward Hacking with Occupancy Measure Regularization"
_ICLR.cc/2024/Conference — Submitted to ICLR 2024_

### Official Review · Reviewer_jdVB · 2023-10-24

**Soundness:** 2 fair
**Presentation:** 2 fair
**Contribution:** 2 fair
**Rating:** 3
**Confidence:** 4

**Summary:**

This paper considers regularizing the learned policy with a safe policy to deal with the reward hacking issue. Instead of constraining the learned policy to the safe policy via minimizing the KL divergence of their action distribution, as existing methods do, authors propose regularizing based on occupancy measure (OM). Theoretical and empirical results show that OM regularization more effectively prevents reward hacking while allowing for performance improvement on top of the safe policy.

**Strengths:**

1. This paper is well written and easy to follow.
2. Preventing reward hacking is an important and interesting topic for RL.

**Weaknesses:**

1. The authors claimed that there are no direct relationships hold for the returns of two policies under any reward and their action distribution divergence. However, Theorem 1 of [1] shows that we can indeed bound the returns of two policies by their action distribution, which from my perspective, just verifies the effectiveness of KL divergence minimization. And I highly recommend authors to thoroughly read [1], which already theoretically and empirically illustrates OM regularization is better than action distribution regularization.
2. Regularizing the learned policy by OM divergence is not a novel idea, and has already been explored in RL, such as [2].

[1] Xu T, Li Z, Yu Y. Error bounds of imitating policies and environments. Advances in Neural Information Processing Systems, 2020, 33: 15737-15749.

[2] Kang B, Jie Z, Feng J. Policy optimization with demonstrations. International conference on machine learning. PMLR, 2018: 2469-2478.

**Questions:**

1. Why would the occupancy measure KL suddenly increases when $\lambda$ increases (Figure 3)? If I am not wrong, as $\lambda$ increases, we focus more on minimizing the OM divergence of the learned policy and the safe policy, so the occupancy measure KL should decrease?
2. What do you mean by "safe" policy? I can not find a formal definition of it. Is it the policy that never visits some unsafe state-action pairs? Then some safe-RL or constrained RL methods should also be compared.
3. Is proposition 3.1 just to conculde that the parameter $\lambda$ is hard to tune if we use action distribution regularization? But in your proposed OM regularization, we need to adversarially train a discriminator, which has been claimed (e.g., [1]) to be unstable and sensitive to hyper-parameters, especially in high-dimensional domains.
4. During training, do we need to roll out the safe policy? If so, OM regularization may need more environment interactions than action distribution regularization.

[1] Becker E, Pandit P, Rangan S, et al. Instability and local minima in GAN training with kernel discriminators. Advances in Neural Information Processing Systems, 2022, 35: 20300-20312.

---

> ### Author Response · Authors · 2023-11-18
> **Response to Reviewer jdVB (1/2)**
>
> We are glad that the reviewers found our paper easy to read and that they believe the reward hacking problem is of significance. Below, we respond to the reviewer’s comments about our theory and algorithm's novelty:
>
> * **Novelty of our theoretical results:** We would like to emphasize that our theoretical findings are unique in their application to the difficult *reward hacking* setup. We acknowledge that there has been some previous work related to the result we show in Proposition 3.2, which bounds the difference between returns of policies in terms of their distance from each other in occupancy measure space [1, 2]. However, we do believe that the result proved in Proposition 3.1 is novel and significant. This result in particular highlights the challenges of using action distribution KL for regularizing goal-oriented agent activity to avoid reward hacking, and to the best of our knowledge, it has not appeared in the literature before. As the reviewer notes, Theorem 1 of Xu et al. does show the result that you can bound difference in returns by action distribution divergences, but this has an additional $\frac{1}{1-\gamma}$ term compared to our bound. Thus, as $\gamma$ grows closer to 1, the previously proved bound becomes less tight. In contrast, our result in Proposition 3.1 utilizes an arbitrarily large $\gamma$ to show that it is impossible to bound the difference in returns using action distribution divergences in a manner that does not depend on the effective horizon $\frac{1}{1-\gamma}$. This is particularly important in long-horizon environments, such as the ones we explore in our experiments. We will be more explicit about which of our theoretical results are similar to those that have appeared in the literature previously and which are novel in the final version of the paper.
>
> * **Clarifying Proposition 3.1:** Proposition 3.1 actually does not claim that $\lambda$ is hard to tune—it instead shows that *no value of $\lambda$ will work for regularization.* Proposition 3.1 shows that arbitrarily small changes in action distributions between policies can cause arbitrarily large differences in return, and vice versa: arbitrarily large changes in action distributions between policies can cause arbitrarily small differences in return. Due to this result, there is a contradiction that exists in selecting the $\lambda$ value: if it is high, then it will prevent any improvement on the safe policy, but if it is low, then it will allow reward hacking to occur, rendering it impossible to regularize effectively using this method.
>
> * **Novelty of ORPO:** We acknowledge that occupancy measure matching has been utilized previously in RL for different applications, and as the reviewer notes, our method of keeping the learned policy within a certain distance in occupancy measure space of the safe policy is similar to the approach proposed by Kang et al. However, we are applying our algorithm to quite a different setting. In particular, in addition to empowering the agent to explore more like in previous works, ORPO is also able to prevent potentially unsafe reward hacking. Additionally, we also consider a suite of more realistic settings when evaluating our method and find that it can perform well in complex environments with both continuous and discrete action spaces.

---

> ### Author Response · Authors · 2023-11-18
> **Response to Reviewer jdVB (2/2)**
>
> We further respond to the reviewer's comments about our experiments and method:
>
> * **OM KL increasing as $\lambda$ increases:** We sometimes see instability at high values of $\lambda$ because the added rewards from the discriminator are oftentimes very large compared to the underlying rewards from the environment, and they are changing over time. As we show in Figure 3, ORPO works best with more moderate values of $\lambda$ since moderate regularization allows for both reward hacking prevention and improvement upon the provided safe policy. So, we do not believe that instability with high values of $\lambda$ presents a challenge to the practical adoption of our regularization method.
>
> * **Definition of a safe policy:** A safe policy is any policy which has reasonable, potentially quite suboptimal, performance and does not reward hack. The objective of ORPO is to stay close enough to the safe policy such that unsafe, reward hacking behaviors can appropriately be avoided, while also allowing for enough exploration to actually perform better than the safe policy. Safe policies can be hardcoded or learned from human data. For instance, as noted in section 4, we use the Intelligent Driver Model, which is a widely accepted approximation of human drivers, as the safe policy for the traffic environment and for the glucose environment, we use glucose values that were selected through rigorous scientific testing. Section 2 of our paper presents several comparisons to previous safe and constrained RL methods.
>
> * **Generating rollouts for the safe policy:** ORPO does require more interaction with the environment because it needs to sample data from both the safe policy and the policy currently being learned in order to train the discriminator. However, because the safe policy is not being updated when ORPO is trained, we can collect rollouts from it at the beginning of training and store it in a replay buffer. This way, we will incur a single initial sampling cost that is amortized over the course of the training process, as these pre-collected samples are utilized throughout the algorithm's iterations.
>
> Please let us know if there are any other comments or questions we can address.
>
> References:
>
> [1] Xu et al. "Error Bounds of Imitating Policies and Environments". NeurIPS 2020.
>
> [2] Yong et al. "What is Essential for Unseen Goal Generalization of Offline Goal-conditioned RL?". ICML 2023.

---

### Official Review · Reviewer_y14X · 2023-11-01

**Soundness:** 2 fair
**Presentation:** 3 good
**Contribution:** 1 poor
**Rating:** 5
**Confidence:** 4

**Summary:**

- The authors defined reward hacking as a problem where an AI agent seems to do well based on a chosen reward function but actually performs poorly according to the actual desired reward.
- Previous solutions have tried to align the proxy reward function with the true reward by matching action distributions using measures like KL divergence, but this can be risky because even a small change in actions at one state can lead to very different state occupancies.
- Authors propose a new approach called ORPO, which regularizes state occupancy measure instead of action distribution, which is more effective in preventing reward hacking while still allowing for performance improvement beyond a safe policy.
- The authors show a theoretical connection between the returns of two policies and their OM divergence, which doesn't exist for action distribution divergence.

**Strengths:**

- The paper is easy to follow and well presented
- Discusses interesting topic of reward hacking, which the authors clearly defined as the "reward mismatch". Previously the reward hacking  problem was somewhat vague.
- Proposition 3.1 shows a nice differentiation between action dist. regul. and om regul.
- experiment shows the superiority of proposed method against action regularization

**Weaknesses:**

- While the paper discusses with a new problem formulation on reward hacking, the problem definition itself does not seem to be very different from what offline RL algorithms do, which assume a transition/reward function mismatch and learns a robust policy for underlying true transition/reward function. The potential differences could be, e.g. surrogate reward function can be largely different, or transition function stays exact, etc. However, the authors do not make much difference in designing their algorithm, and I could not find much difference in proposed OM algorithm and other offline RL algorithms out there.
- In that sense, there are a number of occupancy matching algorithms (algaeDICE, OptiDICE, ...) for offline RL, and I believe that it should have been shown how the proposed OM algorithm is different to those and why it is better suited in this case.

**Questions:**

- In offline RL, having KL-divergence regularization usually results in an over-regularization, since we could have more flexible learned policy by having other divergences that matches support only. Is there any specific reason why KL divergence is chosen in this paper? The similar mathematical results should be able to be derived using many other divergences.

---

> ### Author Response · Authors · 2023-11-18
> **Response to Reviewer y14X**
>
> We thank the reviewer for their interest in our problem definition and their appreciation of our theoretical and empirical results. The reviewer primarily challenged our algorithm's novelty by making comparisons to offline RL methods; however, we argue below that the motivation behind and algorithms for offline RL are fundamentally different from ORPO:
>
> * **Difference between our motivation and that of offline RL:** Offline RL aims to avoid issues from having a state distribution mismatch between the training data available and the states that will be seen during deployment. In contrast, we are trying to solve a problem of when there is a *reward* function mismatch. Specifically, offline RL doesn’t necessarily consider any reward function, and even with knowledge of the environment’s transition dynamics or infinite amounts of data, offline RL algorithms can still perform horribly without any ground truth reward signal, resulting in catastrophic outcomes [1]. Several previous offline RL theoretical results have only provided performance guarantees in the case of when the dataset actually reflects the true reward function [2]. The limitations that are addressed by offline RL methods are also separate from the problem of reward hacking that ORPO addresses. In particular, in the offline RL setting, we will practically have limited amounts of data available, whereas in our setting, we are challenged by a misspecified or “hackable” reward that can motivate unsafe behavior from the agent.
>
> * **Difference between ORPO and offline RL algorithms:**  Offline RL algorithms typically optimize over the empirical transition or reward function found within the provided dataset, which is subject to estimation errors due to the limited amount of data practically available. So far, the approach to account for this estimation error has been to act pessimistically, applying different kinds of reward penalties based on the error [3]; however, this pessimism can result in suboptimal policies that do not explore enough [4].  We acknowledge the fact the occupancy measures have also been used previously in the literature; however, again, ORPO is unique in its emphasis on preventing *reward hacking*. The reviewer in particular mentions algaeDICE, OptiDICE, and other methods from the DICE family [5], which have a dual objective of estimating the ratio between the occupancy measures of both the expected optimal policy and the policy under which the dataset was collected and optimizing the learned policy so that the ratio previously calculated is minimized. These methods do not actually calculate occupancy measures of a particular policy; instead, they use a duality trick to approximate the corrections that will be needed to turn the offline dataset distribution into the expected optimal policy’s distribution. On the other hand, we actually calculate the occupancy measures using a discriminator and incorporate them into our algorithm as a crucial value for regularizing the learned policy to a provided safe policy, rather than trying to approximate some desired policy and adjusting the safe policy’s distribution to match that. We will include a thorough comparison to these and other related methods in the final paper.
>
> * **Using other divergences for ORPO:** After  experimenting with regularization based on the TV and Wasserstein distances between policy occupancy measures, we found them to be much more unstable compared to KL-based regularization. KL divergences are generally favorable since they tend to be more stable to calculate and are easier to approximate using various tricks, such as discriminators. Current state-of-the-art methods also tend to rely on KL divergences for regularizing to safe policies [6, 7, 8, 9], which also contributed to us choosing it for ORPO.
>
> Please let us know if there are any additional questions or comments we can address.
>
> References:
>
> [1] Haoyang He. "A Survey on Offline Model-Based Reinforcement
> Learning". 2023.
>
> [2] Cheng et al. "Adversarially trained actor critic for offline reinforcement learning". ICML 2022.
>
> [3] Rashidinejad et al. "Bridging Offline Reinforcement Learning and Imitation Learning: A Tale of Pessimism". NeurIPS 2021.
>
> [4] Xie et al. "Bellman-consistent Pessimism for Offline Reinforcement Learning". NeurIPS 2021.
>
> [5] Lee et al. "COptiDICE: Offline Constrained Reinforcement Learning via Stationary Distribution Correction Estimation". ICLR 2022.
>
> [6] Nair et al. "AWAC: Accelerating Online Reinforcement Learning with Offline Datasets". CoRL 2019.
>
> [7] Siegel et al. "Keep Doing What Worked: Behavioral Modelling Priors for Offline Reinforcement Learning". ICLR 2020.
>
> [8] Wu et al. "Behavior Regularized Offline Reinforcement Learning". 2019.
>
> [9] Ouyang et al. “Training language models to follow instructions with human feedback”. 2022.

---

### Official Review · Reviewer_Jr8S · 2023-11-01

**Soundness:** 3 good
**Presentation:** 4 excellent
**Contribution:** 2 fair
**Rating:** 6
**Confidence:** 4

**Summary:**

The authors present an alternative method for regularization to a safe policy, via occupancy measure rather than KL divergence of action distribution. The paper explains that action distribution KL divergence is a bad measure of policy difference with respect to reward hacking and resulting behavior because small changes in the action distribution can result in drastically different behavior and vice versa (large differences in action distribution may not matter). State-action (or sometimes state only) occupancy measure is justified as a good measure of alignment between policies both theoretically and empirically. The authors prove that the occupancy measure total variation distance tightly bounds the returns between the policies. They then provide intuition in a small environment and show the difference between action distribution and occupancy measure when comparing the desired policy, a safe policy, and a reward hacking policy. They introduce a PPO-based algorithm to achieve this regularization. Crucially, it is intractable to compute the ground truth TV distance so it* is estimated with a discriminator akin to the one used in GAIL. *Actually they use KL divergence rather than TV distance in practice because it is more stable to compute in practice, and they cite theoretical justification for this trick. Empirical results are shown in a small gridworld, Flow, and a glucose monitoring environment.

**Strengths:**

The idea of regularizing to a safe policy via occupancy measure rather than KL divergence over action distribution is novel. (Although I am somewhat surprised this is the case! I was aware of safe policy regularization as a way to reduce reward hacking, but not familiar with the specific methods, so I'm surprised to learn this.)

The theoretical results appear strong and sound; a tight bound is given relating changes in TV distance of occupancy measure to changes in the true reward. Theoretical justification is provided for why KL divergence is used in practice rather than TV distance, supporting an empirical finding that it is more stable. Furthermore, a theoretical result from Huszar (2017) is cited to connect a proof relating the scores of a GAN discriminator of safe vs learned policy to the KL divergence between occupancy measure. This keen application of discriminators is used to give a tractable algorithm for occupancy measure regularization.

A very simple MDP is provided to give intuition for why action distribution is not a good measure to regularize for.

The gridworld environment provides a clear illustration of the benefits of occupancy measure vs action distribution regularization. Experiments in more complex environments (flow and simglucose) are provided.

The paper is written very clearly and it was easy to follow with minimal effort.

**Weaknesses:**

Overall the paper is strong theoretically and empirically and convinces me of a real issue with the action distribution regularization SOTA. While alignment is important and this work demonstrates a better way to regularize to a safe policy, the novelty is primarily in a detail of that regularization, and I'm surprised people hadn't considered other metrics to regularize besides action distribution KL divergence.

The authors do not address the question of whether occupancy measure regularization is stable / will converge, which is a crucial consideration of the method. I'd like to see some discussion on that both theoretically and empirically. The authors don't show the training curves either (performance over time) so I'm unable to assess convergence.

I'd also like to see some discussion of what sort of environments the authors expect the method to do well in, extrapolating from the experiments. I.e. is this *the* solution to reward hacking or should it be used in conjunction with other methods, and is it effective anywhere or only in environments with particular properties?

**Questions:**

- Why do you choose TV distance in (3) in the first place? is it because it makes the theory resulting in the tight bound nice?
- Can you provide intuition for why the Huszar (2017) result applies to KL divergence rather than TV distance?
- Can you provide empirical comparison of KL div and TV dist occupancy measure regularization?
- How many trials were run? Can you provide error bars for the graphs and ranges for the table?
- Can you analyze the computation complexity (theoretical and empirical)? I guess it shouldn't be too new since it's application of an existing discriminator method.
- In figure 3: how come at the very right end of the plots (high lambda) the occupancy measure KL div increases at the end for glucose? why is performance so poor in traffic and glucose for low lambda? is it due to reward hacking or poor policy optimization? can you plot training reward in addition to what I'm assuming is test/true reward to demonstrate reward hacking?

---

> ### Author Response · Authors · 2023-11-18
> **Response to Reviewer Jr8S (1/2)**
>
> We are glad that the reviewer appreciated our paper's presentation, and that they found our results strong and clearly communicated. Below we respond to their comments and criticisms:
>
> * **Stability of ORPO:** The reviewer asked if occupancy measure regularization is stable and will converge. Deep RL is poorly understood theoretically [1, 2], and furthermore, it has been generally difficult to make theoretical progress on other algorithms that use a discriminator, like GANs or GAIL. Thus, it is very difficult to theoretically analyze the stability and convergence of ORPO. However, in practice, we find that training is stable. Please reference Appendix B, Figure 4 to see learning curves that show the true reward, proxy reward, and estimated occupancy measure KL over time for ORPO in each environment. In this graph, the training is relatively stable and appears to converge. We do sometimes see instability at higher values of $\lambda$ because the added rewards from the discriminator are very large compared to the underlying rewards and are changing over time. This explains why the occupancy measure divergence for glucose appears to be increasing for large values of $\lambda$. As we show in Figure 3, ORPO works best with moderate values of $\lambda$ since moderate regularization both prevents reward hacking and allows for improvement upon the provided safe policy. Thus, we do not believe that instability with high values of $\lambda$ presents a challenge to the practical adoption of our regularization method.
>
> * **Applicability to a diverse set of environments:** The reviewer also asked about which environments we expect ORPO to be most useful in. We anticipate ORPO to be broadly applicable across a wide range of realistic environments since our theoretical results suggest that occupancy measure KL is an effective regularizer for **any** environment and **any** reward function. While our approach is a crucial step towards preventing reward hacking, we do not believe that it, as a regularization method, can single-handedly address all challenges posed by reward misspecification. Ideally, our algorithm should be combined with methods to improve the reward function and/or to monitor anomalous behaviors. However, we believe that ORPO can be a key part of preventing reward hacking and should replace action distribution regularization as the standard method for regularizing to a safe policy.

---

> ### Author Response · Authors · 2023-11-18
> **Response to Reviewer Jr8S (2/2)**
>
> We also respond to the questions posed by the reviewer:
>
> * **The choice of TV over KL in our theoretical results:** When presenting our theoretical results, we choose the TV distance as it has nice theoretical properties that result in the tight bound we find. It is also preferable for our proofs since its magnitude is bounded. However, as stated at the start of Section 3.4 in our paper, we rely on the KL divergence within our algorithm ORPO since it is more stable to calculate in practice. Furthermore, because Pinsker's inequality bounds the TV distance in terms of the KL divergence, the nice theoretical properties we find for the TV distance between the occupancy measures of policies also hold for the KL divergence. Huszar (2017) used KL divergence because of its relevance to the Variational Inference literature. Specifically, KL is always differentiable, which can be useful when training structures such as GANs, whereas TV isn't always differentiable everywhere. In addition, KL divergence’s asymmetry is actually seen as a desirable quality since it allows for the variable overestimation and underestimation of probability in different parts of the distributions.
> * **Empirical comparison of KL and TV:** We experimented with regularization based on the TV distance between policy occupancy measures but found it to be much more unstable. Current state-of-the-art methods RL methods also tend to rely on KL divergences for regularizing policies instead of TV distance since they have more stability and can be approximated more easily [3, 4, 5, 6], which also contributed to us choosing it for ORPO.
>
> * **Our experimentation process:** We have done some more experimentation, increasing the coverage of $\lambda$ values. Figure 3 and Table 1 in the paper have been updated with the new results, along with error bars and metric standard deviations.  In some cases, the variance across seeds is quite high, oftentimes due to the general instability associated with RL training, but in the best-performing cases, the variance is relatively lower and shows that ORPO outperforms action distribution regularization. For instance, in the case of the glucose and traffic environments, even the lowest performing seed of the best coefficient for occupancy measure regularization outperforms the best performing seed of the best coefficient for action distribution regularization.
> * **Computational complexity:** Most policy gradient algorithms like PPO are trained online, and as a result, a key contributor to their computational complexity is generally sampling trajectories from the environment on each policy update [7]. ORPO does require more interaction with the environment because it needs to sample data from both the safe policy and the policy currently being learned in order to train the discriminator. However, because the safe policy is not being updated when ORPO is trained, we can collect rollouts from it at the beginning of training and store it in a replay buffer. This way, we will incur a single initial sampling cost, which are amortized over the course of the training process, as these pre-collected samples are utilized throughout the algorithm's iterations. ORPO also requires the training of a discriminator, but since we use relatively small networks for the discriminator, we found that this added negligible time.
>
> * **Comments about specific results:** As we noted above, OM KL calculations can be more unstable at higher values of $\lambda$, which is why the OM divergence increases in glucose for larger $\lambda$ values. We find that large values of $\lambda$ over-regularize and are too unstable, and smaller $\lambda$ values do not regularize enough to prevent reward hacking, resulting in poor performance in all of the environments. Please reference the traffic environment plots in Appendix B, Figure 4 for a clear example of learning curves that show an initial spike in proxy reward and a dip in true reward, followed by an increase in true reward as the policy is increasingly regularized.
>
> Please let us know if there are any other comments or questions we can address.
>
> References:
>
> [1] Lin et al. "Why does deep and cheap learning work so well?". *Journal of Statistical Physics* 2017.
>
> [2] Laidlaw et al. "Bridging Reinforcement Learning Theory and Practice with the Effective Horizon". NeurIPS 2023.
>
> [3] Nair et al. "AWAC: Accelerating Online Reinforcement Learning with Offline Datasets". CoRL 2019.
>
> [4] Siegel et al. "Keep Doing What Worked: Behavioral Modelling Priors for Offline Reinforcement Learning". ICLR 2020.
>
> [5] Wu et al. "Behavior Regularized Offline Reinforcement Learning". 2019.
>
> [6] Ouyang et al. “Training language models to follow instructions with human feedback”. 2022.
>
> [7] Meng et al. "Off-policy proximal policy optimization". AAAI 2023.

---

> > ### Comment · Reviewer_Jr8S · 2023-11-23
> >
> > Thank you for the detailed responses to my questions and comments, they have been well addressed.

---

### Official Review · Reviewer_M4hZ · 2023-11-02

**Soundness:** 3 good
**Presentation:** 3 good
**Contribution:** 2 fair
**Rating:** 6
**Confidence:** 4

**Summary:**

This paper considers the problem of reward hacking in reinforcement learning (RL), where an RL is trained to optimize a proxy reward that can lead to suboptimal performance on the true reward. Given a safe policy, $\pi_{safe}$, that performs adequately with respect to the true reward, a common approach in the literature is to train a policy, $\pi$, to optimize the proxy reward while trying to ensure that the KL-divergence $D_{KL}(\pi(\cdot | s) \ || \ \pi_{safe}(\cdot | s))$ , or *action distribution divergence*, is small at each $s$. The paper provides examples illustrating the drawbacks of this approach and proposes regularizing using divergences between state(-action) occupancy measures, instead. The primary contributions are:
1) theoretical results that (a) illustrate when action distribution divergence regularization fails, and (b) draw connections between suboptimality due to reward hacking and occupancy measure divergence regularization;
2) a practical method for performing occupancy measure divergence-regularized policy optimization;
3) experimental results illustrating benefits of the proposed approach on three environments.

**Strengths:**

The paper proposes an interesting way to deal with reward hacking that is likely of interest to the community. This is based on the useful insight that, in problems where the state space has a clear geometric interpretation or where rewards are purely state-dependent, encouraging the policies trained to remain within a "safe" region prescribed by an existing policy can help prevent reward hacking in the case where the proxy reward differs greatly from the true reward outside that safe region. The overall motivation is very clear and convincing, and the illustrative examples presented (the car example in the intro, two-state example in Fig. 1, and gridworld environment pictured in Fig. 2) are effective. The proposed ORPO algorithm is interesting, and the experiments support its effectiveness on three environments.

**Weaknesses:**

Though the key idea and motivation of the paper is nice, there are important weaknesses, including:
1. The paper claims Proposition 3.2 (page 5) as a primary theoretical contribution, yet its proof in the appendix is not novel: the supporting Prop. A.1 is standard in MDP theory, as the RHS is the objective of the dual LP formulation for solving an MDP; the remaining bound is trivial (it is also previously known -- see its use in, e.g., the proof of Theorem 1 in [Qu et al., *Scalable multi-agent reinforcement learning for networked systems with average reward*, NeurIPS 2020]). This undermines the theoretical contribution of the paper.
2. The idea of using state(-action) occupancy measure divergence to address reward hacking does appear to be novel, yet using it as part of an RL objective has been fairly extensively studied -- see, e.g., [Hazan et al. *Provably efficient maximum entropy exploration*, ICML 2019], [Lee et al., *Efficient exploration via state marginal matching*, arXiv 2019], [Liu & Abbeel, *Behavior from the void: unsupervised active pre-training*, NeurIPS 2021]. It would be helpful to see the proposed method put in context of these and related works to more clearly highlight its novelty and the paper's methodological contribution.
3. The experiments are not as extensive as one could hope. It is noted in the appendix that three replications are performed for each method, but only the median rewards are reported -- without error bars (and ideally more replications), it is tough to assess the statistical significance of the results reported in Table 1 and Fig. 3. In addition, though hyperparameters are reported, it is unclear how they were selected to ensure a fair comparison between all methods. In light of issue 1, it would be helpful to have more thorough experimentation to address 3.

**Questions:**

Specific questions:
* how significant are the examples constructed in Prop. 3.1? these seem to require some work in the appendix -- do you view Prop. 3.1 as a significant contribution on its own?
* what are the variance/confidence intervals for the values reported in Table 1? what about Fig. 3?
* the true reward plots in Fig. 3 are a little confusing -- how should we interpret them?
* why are some of the KL plots truncated in the Tomato experiment in Fig. 3?

---

> ### Author Response · Authors · 2023-11-18
> **Response to Reviewer M4hZ (1/2)**
>
> We thank the reviewer for their comprehensive feedback. We are glad that the reviewer found our argument for occupancy measure-based regularization clear and convincing. Below is our response to the criticisms and questions raised about the theory and algorithm that we presented:
>   * **Significance of theoretical results:** We would like to emphasize that our theoretical findings are unique in their application to the *reward hacking* setup. We do acknowledge that there has been some previous work related to the result we show in Proposition 3.2, which bounds the difference between returns of policies in terms of their distance from each other in occupancy measure space [1, 2]; however, we believe that the result we prove in Proposition 3.1 is novel and significant. Proposition 3.1 shows that arbitrarily small changes in action distributions between policies can cause arbitrarily large differences in return, and vice versa: arbitrarily large changes in action distributions between policies can cause arbitrarily small differences in return. This result in particular highlights the challenges of using action distribution KL for regularizing goal-oriented agent activity to avoid reward hacking, and to the best of our knowledge, it has not appeared in the literature before. We will be more clear about which of our theoretical results are similar to those that have appeared in the literature previously and which are novel in the final version of the paper.
> * **Occupancy measures for RL:**  We also acknowledge that occupancy measures (OM) have been studied and utilized in the literature previously for imitation learning, offline RL, and efficient exploration [3, 4, 5]; however, our usage of OM-based divergence for policy regularization to resolve the relatively unexplored, yet critical problem of *reward hacking* is what makes our method novel. The reviewer in particular mentions methods for maximum entropy exploration, which optimize for a lower bound of the policy-induced steady-state distribution's entropy that can then be used to define intrinsic rewards [6,7]. Our method is fundamentally different as we are trying to regularize the behavior of the agent so that it is not only safe but also an improvement with respect to the provided safe policy, **not** artificially construct rewards for under-specified environments. Other similar works, such as state marginal matching, assume that the system designer has some knowledge about the target distribution to which the learned policy's state distribution must be aligned [8], whereas our algorithm requires no extra input other than a reasonably specifiable policy that doesn't exhibit reward hacking behaviors. While these methods provide an effective way to reconcile with the exploration-exploitation trade-off, they fail to guarantee the safety of the agent as it is still reasonable to expect that even with this principled approach towards exploration, the agent can find ways to hack the specified goal, since there is nothing preventing the discovery of these dangerous states. We will better address these and other related literature in the final version of our paper.

---

> ### Author Response · Authors · 2023-11-18
> **Response to Reviewer M4hZ (2/2)**
>
> We also respond to comments made about our experimentation process:
> * **More Robust Experimentation:** We have done some more experimentation, increasing the coverage of $\lambda$ values. Figure 3 and Table 1 in the paper have been updated with the new results, along with error bars and metric standard deviations. In some cases, the variance across seeds is quite high, oftentimes due to the general instability associated with RL training, but in the best-performing cases, the variance is relatively lower and shows that ORPO outperforms action distribution regularization. For instance, in the case of the glucose and traffic environments, even the lowest performing seed of the best coefficient for occupancy measure regularization outperforms the best performing seed of the best coefficient for action distribution regularization.
>
> * **Hyperparameter Selection:** As noted in Appendix D.1, we largely used the hyperparameters for all environments from Pan et al. In some cases, we adjusted hyperparameter values to ensure that reward hacking consistently occurred when an unregularized RL agent (i.e., PPO without regularization) was taking actions in the environment, so that we could clearly compare methods for preventing this reward hacking. Between action distribution regularization and ORPO we used the same hyperparameters, so we do not believe that our hyperparameter selection advantaged one method over the other.
>
> * **Explanation of Figure 3:** The true reward plots in Figure 3 reflect what happens to the performance of a policy optimizing a misaligned proxy reward function when evaluated using the unknown true reward as we increase the regularization coefficient $\lambda$. With small values of $\lambda$, there is almost no regularization, resulting in reward hacking and thus very low true reward. With moderate values of $\lambda$, there is enough regularization to prevent reward hacking while still allowing for improvement upon the safe policy—the policies with moderate levels of regularization tend to do better than the specified safe policies, whose true rewards are shown as dotted lines. With large values of $\lambda$, there is too much regularization, which results in a policy that performs similarly to the safe policy.
>
> Please let us know if there are any other questions or comments we can address.
>
> References:
>
> [1] Xu et al. "Error Bounds of Imitating Policies and Environments". NeurIPS 2020.
>
> [2] Yong et al. "What is Essential for Unseen Goal Generalization of Offline Goal-conditioned RL?". ICML 2023.
>
> [3] Ho et al. "Generative Adversarial Imitation Learning". NeurIPS 2016.
>
> [4] Mandal et al. "Performative Reinforcement Learning". 2023.
>
> [5] Yang et al. "Regularizing a Model-based Policy Stationary Distribution
> to Stabilize Offline Reinforcement Learning". ICML 2022.
>
> [6] Hazan et al. "Provably Efficient Maximum Entropy Exploration". ICML 2019.
>
> [7] Nedergaard et al. "k-Means Maximum Entropy Exploration". 2022.
>
> [8] Lee et al. "Efficient Exploration via State Marginal Matching". 2019.

---

> > ### Comment · Reviewer_M4hZ · 2023-11-23
> >
> > Thanks to the authors for their response. Understood regarding the previous existence of Prop. 3.2. The additional explanation regarding Prop. 3.1 is helpful, and the examples provided therein do provide interesting support to the overall narrative. For your response to point 2 in the Weaknesses, I understand your point regarding pure exploration, but with respect to state marginal matching, for example, it still seems to me that your formulation of reward hacking might be accommodated by simply taking the policy whose marginal is to be matched in SMM to be a "reasonably specifiable policy that doesn't exhibit reward hacking behaviors". The point is that occupancy measure divergence has been previously used to encourage behavior that remains close to a desirable policy while learning other behaviors -- it still remains unclear whether there is something unique in the reward hacking setting that renders these previous methods inapplicable. Finally, the additional clarity in the experiments is helpful. In light of the authors' response, I have increased my score. I hope to see additional discussion of the significance of Prop. 3.1 and the differences between the proposed approach and previous works using occupancy measure divergences in the final version.

---

### Official Review · Reviewer_rjUA · 2023-11-05

**Soundness:** 3 good
**Presentation:** 3 good
**Contribution:** 2 fair
**Rating:** 5
**Confidence:** 2

**Summary:**

This paper proposes a new method for preventing reward hacking in reinforcement learning. It highlights the limitations of the commonly used KL divergence between action distributions for regularization and proposes occupancy measure (OM) as a more effective alternative. Specifically, the paper identifies the limitations of action distribution based regularization where the authors show how small changes in action distribution can lead to significantly different outcomes, potentially causing calamitous results. Moreover, the paper introduces occupancy measure-based regularization: occupancy measure captures the proportion of time an agent spends in each state-action pair, which provides a more comprehensive picture of the policy's behavior. Furthermore, the paper provides theoretical justification of the approach via proving a direct relationship between the returns of two policies under any reward function and their OM divergence, demonstrating the effectiveness of OM for reward hacking prevention. Finally, the paper demonstrates through experiments that OM regularization outperforms action distribution-based methods in preventing reward hacking while allowing for performance improvements.
This work suggests that OM regularization is a promising technique for safe and effective reinforcement learning, offering advantages over traditional action distribution-based methods in mitigating reward hacking.

**Strengths:**

The paper presents a reasonable approach that mitigates reward hacking via occupancy measure matching in a way that is easy to follow and understand. Moreover, the paper shows a simple toy example to demonstrate the importance of using occupancy measure matching instead of action distribution matching, which makes it clear for readers to understand.

The paper also shows theoretical justifications of why using KL divergence between safe policy and learned policy might not be ideal for mitigating reward hacking and why occupancy measure matching is better, which makes the paper more principle.

Finally, the authors provide empirical results that aligns with the theoretical finding and strengthen the paper a lot.

**Weaknesses:**

1. While the idea of the paper is neat, it is not super novel as similar ideas have been explored in previous works such as GAIL and [1]. It is unclear if the theoretical finding in the paper is fully new as it has been known in the literature that KL divergence as the action distribution regularization may not be ideal and occupancy measure matching can capture the total return variation more accurately [2].

2. Moreover, the empirical results in the paper are less realistic. The experiments are all in simulated environment with not very complex control space. It would add more value if the authors can perform experiments in more complicated settings such as continuous-control settings like robotic manipulation/locomotion or large language models alignment setting that require RLHF.

[1] Ma, Yecheng Jason, Kausik Sivakumar, Osbert Bastani, and Dinesh Jayaraman. "Policy Aware Model Learning via Transition Occupancy Matching." In Deep Reinforcement Learning Workshop NeurIPS 2022. 2022.

[2] Wu, Yueh-Hua, Ting-Han Fan, Peter J. Ramadge, and Hao Su. "Model imitation for model-based reinforcement learning." arXiv preprint arXiv:1909.11821 (2019).

**Questions:**

1. Please clarify the novelty of the paper.
2. Please justify the new findings in the theoretical analysis that differs from previous works.
3. Perform more realistic tasks such as LLM RLHF tasks and complex robotic manipulation/locomotion.

---

> ### Author Response · Authors · 2023-11-18
> **Response to Reviewer rjUA (1/2)**
>
> We thank the reviewer for their commentary. We appreciate the fact that they thought our paper was easy to follow and found the empirical evidence well-aligned with the theoretical findings presented. Below is our response to the criticisms and questions raised:
>
>  * **Novelty of using occupancy measures:** We acknowledge the fact that occupancy measure (OM) matching has been studied and utilized in the literature previously; however, our usage of OM divergence for policy regularization to resolve the relatively unexplored, yet critical problem of *reward hacking* is what makes our method novel. The reviewer in particular mentioned GAIL [1]. While GAIL does also rely on state-action occupancy measures, similar to our method, it is primarily focused on finding a policy that replicates as expert’s demonstrations. In contrast, our method ORPO is focused on *improving* beyond a safe policy (e.g., one trained via imitation learning). Because ORPO combines occupancy measure divergence and reward objectives, it can both outperform the safe policy and avoid reward hacking. We will include a detailed comparison to GAIL and other prior works that use occupancy measure divergences in the final paper if accepted.
>  * **Novelty of theoretical analysis:** We also acknowledge that there has been previous work that has focused on bounding, e.g., the difference in value functions between policies whose occupancy measure divergence is bounded [2]—this is quite similar to our Proposition 3.2. However, we believe our Proposition 3.1 is novel. It shows that arbitrarily small changes in action distributions between policies can cause arbitrarily large differences in return, and vice versa: arbitrarily large changes in action distributions between policies can cause arbitrarily small differences in return. This result in particular highlights the challenges of using action distribution KL for regularizing RL to avoid reward hacking, and to the best of our knowledge has not appeared in the literature before. We will be more clear about which of our theoretical results are similar to those which have appeared in the literature before and which are novel in the final version of the paper.
>  * **How realistic are our experiments:** The reviewer mentioned that they feel our experiments are less realistic and suggested evaluating on robotics domains. However, we argue that the diverse environments used in our paper for experimentation are actually highly realistic, with the exception of the toy tomato-watering gridworld environment. In general, we believe that these settings are better for robustly evaluating ORPO than robotics tasks. The traffic simulation and glucose monitoring environments have been crafted by domain experts and are backed by extensive scientific studies. As we note in Section 4, the Flow traffic simulator is based on SUMO [3], which was developed by members of the Institute of Transportation Systems. It has been used to study various complex real-world scenarios related to mixed-autonomy traffic and the practical deployment of autonomous vehicles [4,5]. In addition, the safe policy for the traffic environment is the Intelligent Driver Model that is widely accepted by professionals in the field [6]. The SimGlucose Type 1 Diabetes monitoring was built off of the FDA-approved UVa/Padova Simulator [7], and the blood glucose and other patient state values used in the simulated scenarios have been calibrated by leading diabetes researchers from the Harvard Medical School and Boston Children's Hospital [8]. Unlike many common robotics benchmarks, our environments require balancing objectives that are difficult to specify, such as “good traffic” and “patient health/financial wellness”; this leads to naive reward functions being easily hackable, as we find in our experiments. In contrast, most common robotics benchmarks, like DM Control, have relatively simple reward functions that cannot be hacked, so it is more difficult to evaluate ORPO in those.
>
> Overall, we argue that our theory and experiments are a significant contribution. Our lower bounds in Proposition 3.1 show that action distribution regularization can perform arbitrarily poorly, and our experiments on realistic environments demonstrate that our theory translates to practical benefits for regularized reinforcement learning.

---

> ### Author Response · Authors · 2023-11-18
> **Response to Reviewer rjUA (2/2)**
>
> **Note about Experiments on LLM RLHF:** The reviewer also suggested we perform experiments on regularizing RL as part of RLHF for large language models (LLMs). In this case, action distribution-based policy regularization is widely used [9]. It turns out that for LLMs, action distribution regularization is actually *equivalent* to occupancy measure regularization. The occupancy measure $\mu$ of an LLM's state $s$, the response generated until timestep $t$ using words $w_1, w_2,...w_t$, can be written in terms of the autoregressive model $\pi$'s predictions:
>    $$\log \mu(s) = \log \pi(w_t|w_{t-1},...w_1) + \log \pi(w_{t-1}|w_{t-2},...w_1) + \log \pi(w_1)$$
>
>    Here, the next word given the previous sequence is the action $a$ to be taken. Thus, the KL divergence $D_\text{KL}$ between the SFT model $\pi_{safe}$ and the RLHF model $\pi$ and the KL divergence between their corresponding occupancy-measures, $\mu_\text{safe}$ and $\mu$ are roughly equivalent:
>    $$D_\text{KL}(\mu \| \mu_\text{safe}) = \sum_{s,a}\mu(s,a) * \sum_t [\log\mu(s,a) - \log\mu_\text{safe}(s,a)]$$
>    Combining the information from the two equations above, we get the desired result:
>    $$D_\text{KL}(\mu \| \mu_\text{safe}) = D_\text{KL}(\pi \mid \pi_\text{safe})$$
>    Therefore, while we show that our method is very valuable for regularizing the behavior of agents trained with reinforcement learning in general, for LLMs in particular it is equivalent to the already widespread practice of action distribution regularization. This is why we do not evaluate ORPO for RLHF of LLMs. We will include these results showing why ORPO is not necessarily applicable to LLM training in the final version of the paper.
>
> Please let us know if there are any other remaining questions or comments that we can help address.
>
> References:
>
> [1] Ho et al. "Generative Adversarial Imitation Learning". NeurIPS 2016.
>
> [1] Xu et al. "Error Bounds of Imitating Policies and Environments". NeurIPS 2020.
>
> [3] Simulation of Urban Mobility (SUMO) User Documentation
>
> [4] Vinitsky et al. “Optimizing Mixed Autonomy Traffic Flow With Decentralized Autonomous Vehicles and Multi-Agent RL”. *ACM Transactions on Cyber-Physical Systems* 2020.
>
> [5] Cui et al. “Scalable Multiagent Driving Policies For Reducing Traffic Congestion”. AAMAS 2022.
>
> [6] Treiber et al. “Congested Traffic States in Empirical Observations and Microscopic Simulations”. *Physical Review E*, 62(2):1805–1824, 2000.
>
> [7] Man et al. “The UVA/PADOVA Type 1 Diabetes Simulator”. *Journal of Diabetes Science and Technology*, 8(1):26–34, 2014.
>
> [8] Garry M. Steil. "Algorithms for a Closed-Loop Artificial Pancreas: The Case for Proportional-Integral-Derivative Control". *Journal of Diabetes Science and Technology*, 7(6):1621–1631, 2013.
>
> [9] Ouyang et al. “Training language models to follow instructions with human feedback”. 2022.

---

### Meta-Review · Area_Chair_uBKN · 2023-12-06

**Metareview:**

The paper investigates the problem of reward hacking in RL settings where an agent optimizing a proxy reward function could lead to poor behavior on the true reward function. The proposed method is based on the idea of regularizing the policy using occupancy measure instead of action distribution. Both theoretical and empirical results are presented to demonstrate the effectiveness of the proposed method. The reviewers acknowledged that the paper studies an important and practical problem of preventing reward hacking in RL. However, the reviewers pointed out several weaknesses in the paper, and raised concerns related to the limited novelty of using occupancy measure regularization and limited empirical evaluation. We want to thank the authors for their detailed responses. Based on the raised concerns and follow-up discussions, unfortunately, the final decision is a rejection. Nevertheless, the reviewers have provided detailed and constructive feedback. We hope the authors can incorporate this feedback when preparing future revisions of the paper.

**Justification For Why Not Higher Score:**

The reviewers pointed out several weaknesses in the paper, and raised concerns related to the limited novelty of using occupancy measure regularization and limited empirical evaluation. A majority of the reviewers support a rejection decision.

**Justification For Why Not Lower Score:**

N/A

---

### Decision · Program_Chairs · 2024-01-16

Reject